# Clinical Evidence of Biomimetic Hydroxyapatite in Oral Care Products for Reducing Dentin Hypersensitivity: An Updated Systematic Review and Meta-Analysis

**DOI:** 10.3390/biomimetics8010023

**Published:** 2023-01-06

**Authors:** Hardy Limeback, Joachim Enax, Frederic Meyer

**Affiliations:** 1Faculty of Dentistry, University of Toronto, Toronto, ON M5G1G6, Canada; 2Dr. Kurt Wolff GmbH & Co. KG, Research Department, Johanneswerkstr. 34-36, 33611 Bielefeld, Germany

**Keywords:** hydroxyapatite, dentin hypersensitivity, remineralization, dentin tubule, quality of life, pain, systematic review, meta-analysis

## Abstract

Dentin hypersensitivity (DH) is a very common dental problem that can have a negative impact on the quality of life and can lead to invasive dental procedures. Prevention of DH and control of symptoms are highly desirable. Hydroxyapatite (HAP) has been shown in vitro to block dentinal tubules and in vivo to be a safe and effective additive in oral care products that reduce DH clinically. This study’s aim was to conduct a systematic review and meta-analysis of the current evidence that HAP-containing oral care products reduce DH. Databases were searched, and only clinical trials in humans were included; studies conducted in vitro or on animals were not included. Publications in a foreign language were translated and included. We found 44 published clinical trials appropriate for systematic analysis. More than half of the trials had high-quality GRADE scores. HAP significantly reduced dentin hypersensitivity compared to placebo (39.5%; CI 95% [48.93; 30.06]), compared to fluoride (23%; CI 95% [34.18; 11.82]), and with a non-significant tendency compared to other desensitizing agents (10.2%; CI 95% [21.76; −19.26]). In conclusion, the meta-analysis showed that HAP added to oral care products is a more effective agent than fluoride in controlling dentin hypersensitivity and may be superior to other desensitizers.

## 1. Introduction

While enamel is acellular and comprised of nearly entirely inorganic mineral apatite, making it relatively inert, healthy teeth can respond to stimuli applied to the tooth such as thermal stimuli (ice and heat) and tactile because the underlying connective tissue, the dentin, is connected to the pulp complex, a highly innervated live tissue with nerve fibers capable of detecting stimuli and transmitting signals to the brain. Teeth become ‘extra’ sensitive for many reasons. For example, hairline cracks, poorly bonded restorations, or advanced carious lesions can cause pathological pain that brings patients into the dental office for treatment. Another very common problem of dental pain is called dentin hypersensitivity (DH), usually the result of exposure to the surface of dentin, generally at the cement-enamel junction. Gingival recession and loss of cementum from excessive tooth brushing or demineralization by highly acidic dietary (or external) acidic challenges can cause the entrances of dentin tubules to be exposed to the oral environment. Once this happens, even the mildest cold or airflow stimuli or focused tactile stimuli can result in sharp, severe pain. DH is, therefore, a “short, sharp pain arising from exposed dentin in response to stimuli, typically thermal, evaporative, tactile, osmotic, or chemical and which cannot be ascribed to any other form of dental defect or disease” [1,2,3]. It is this form of dentin pain that is the focus of this systematic review.

The current accepted theory of the induction of pain from DH is the “hydrodynamic theory” [4,5]. When the extracellular fluid in dentin tubules moves, the dentinal tubule processes of the odontoblasts can detect this movement. Through their close contact with the afferent pain nerve fibers in the tubules, the odontoblasts transmit the pain sensation. The movement of fluid can be created through desiccation, thermal changes (cold hot), physical force (fingernail or dental explorer), or osmotic pressures (dissolution of sugars).

DH is very common in the adult population. A recent systematic review showed that patients of all ages with permanent teeth have reported DH, with a prevalence of about one-third of the population at any given time across all studies [6].

The primary strategy for treating DH and maintaining reduced hypersensitivity is to physically cover the exposed dentin and prevent the movement of fluid in or out of the dentin tubules. Conservative procedures should be considered prior to using irreversible ones, such as dental fillings, or periodontal surgery [2]. Chemically occluding the entrances to the dental tubules, either with in-office procedures or with therapies administered at home, is a more conservative and effective approach. There are several agents that have been investigated in the past to conservatively treat DH [7,8]. There have been 8 systematic reviews of therapeutic treatment of dentin hypersensitivity published in the recent past that included studies on hydroxyapatite (HAP) as an active desensitization agent [8,9,10,11,12,13,14,15] (Appendix A: Appendix A). They included studies on HAP that ranged in number from just one to as many as 20 studies. All of them concluded that HAP was an effective dentin desensitizer. Two systematic reviews concluded that HAP was superior to other methods of controlling dentin hypersensitivity [11,13]. Those treatments that achieve dentin tubule occlusion with physical deposits for extended periods of time are considered better treatments than those that only achieve that on a short-term basis. Ingredients that encourage and speed up the natural remineralization process are also well suited for lowering DH symptoms. Biomimetic HAP seems to be one of those agents that fulfills both roles, since it is very similar to the HAP crystals between and intertwined within the collagen fiber bundles of dentin [16].

Since the most recent meta-analysis on the science of HAP reducing DH is 3 years old and we wanted to include foreign language studies, we conducted an updated systematic review and meta-analysis on all the RCTs where HAP was shown in clinical trials to reduce DH.

## 2. Materials and Methods

We used the PICO framework to guide the focus of this literature review. P: Patients—patients of all ages with healthy, non-carious dentitions with some level of dentin hypersensitivity. Patients undergoing periodontal or vital bleaching were not excluded. I: Intervention—the introduction of one of the following oral care products containing biomimetic HAP as an active ingredient; toothpaste, mouthwash, professional product or gel, either in-office professionally administered or self-administered at home. C: Comparison—no intervention (comparison to baseline), placebo controls (HAP-free oral care products), and positive controls (containing other desensitizing agents) were all considered. O: Outcome—a reduction in dentin hypersensitivity, which included reduction from tactile, cold air, ice water, heat, and electrical stimuli as measured by electric pulp testing, visual analogue scales, ordinal scale scores, or subject questionnaire self-assessments.

The following primary databases were searched: PubMed (Ovid Medline), EMBASE, Scopus, Cochrane Library, and Web of Science. Google Scholar was also searched. Two authors had a previous list of published papers. These provided 3 additional publications found outside of the search. The PRISMA guidelines for literature searches [17] were followed (see Appendix A for the completed PRISMA-S checklist). We did not limit our search to English language publications. We found studies in the Korean, Italian, German, and Russian languages. These were translated using Google Translate. We searched the literature up to and including 1 May 2022. No studies on animals, in vitro or in situ human studies were included, even though mechanistic occlusion of exposed dentin tubules is the proposed mechanism of HAP desensitization. We also crosschecked the references that were reported in our previous comprehensive search [18]. For this updated meta-analysis on the efficacy of HAP in reducing DH, we were interested only in human clinical trials providing clinical evidence of efficacy in patients who could report changes in dentin hypersensitivity.

A qualitative analysis (synthesis) was completed for the studies that met the inclusion criteria. We rated the quality of the evidence using the guidelines in dentistry and GRADE graphics described by Richards et al. [19]. A Cochrane Risk of Bias (RoB) analysis using the methods of Sterne et al. [20] was conducted, and a table was generated.

For the meta-analysis, studies that met the inclusion criteria where at least two groups were compared were used. Where baseline data and data from the final examination were available, we calculated the mean reduction of DH (Schiff-Score, Wong–Baker, or VAS-Scale) for each group and then the difference between the groups. Those data were also calculated as the mean relative difference between the groups (in %), which were then used for the meta-analysis. Three different forest-plots were generated: HAP compared to placebo (1), HAP compared to fluoride (2), and HAP compared to other known actives for reducing DH (3). As (3) comprises many different active ingredients, weighting of the sample sizes was not performed for all analyses (1–3) to reduce the possible risk of bias. The calculation and meta-analysis were performed using the open-source software R, version 4.2.1 (R-project.org). We also used the packages *dplyr* and *forestplot* [21].

## 3. Results

Despite limiting our search to human dentin sensitivity and HAP, nearly 40,000 titles had to be screened in order to avoid missing any published studies. After duplicates and irrelevant papers, reviews, abstracts, book chapters, experiments conducted on animals, in vitro and in situ were all rejected (for not meeting our inclusion criteria), we found 44 relevant clinical trials where HAP was investigated clinically for the reduction of DH. Figure 1 summarizes the results of the search using the strategies outlined in the methods.

A complete list of search words is provided in the Appendix A, along with the results of the number of citations found. The details of the 44 publications that met the inclusion criteria for the qualitative and quantitative synthesis are shown in Table 1. All retrieved studies were read in detail and assessed for quality. The GRADE assignments are shown in the table.

### 3.1. Testing Dentin Hypersensitivity (DH)

A variety of tests to illicit DH were used. The most common was an air blast using compressed air from a standard dental chair air–water syringe. Ice, or ice water, was also used as a cold stimulus, and tactile stimuli were used by applying a dental explorer. Most researchers have standardized their stimuli (same distance, isolation of neighboring teeth). Some studies evaluated DH at baseline, and many evaluated only one time point after the start of the clinical trials. Others tested DH at several time points at 2, 3, or even 4 weeks apart after measuring the baseline DH.

### 3.2. Dentin Hypersensitivity (DH) Scoring Results

Nearly all the studies involved using a patient response scoring system that involved a 4-point scale of increasing sensitivity severity (Schiff score [66]), a 10-point visual analogue scale (VAS), or one that required the subject to place a mark on a distance scale (e.g., 10 cm). One study [49] used an electric pulp tester, which eliminated the subjective aspects of dentin sensitivity reporting. Another study [37], which involved younger patients, used the Wong–Baker FACES pain rating scale [67]. The sensitivity tests were reproducible, accurate, and produced, in nearly all the clinical trials, changes in dentin hypersensitivity that showed statistically significant improvements in comfort in the subjects examined.

### 3.3. Qualitative Synthesis

#### 3.3.1. GRADE Assignments

Of the 44 clinical trials found, half were double-blinded and randomized clinical trials (RCTs). The quality of those RCTs was rated as moderate to high. Some studies reported as RCTs were downgraded because they failed to provide the methods used for randomization. Some claimed they were blinded studies but did not provide the details of how the examiners or patients were blinded. These also received a lower GRADE score. Of the 44 trials, 11 were conducted to investigate the HAP application in an office setting with one or two applications. The others involved sending the subjects home with products to use. The length of the studies varied from a few days to 3 months. One study was conducted for 6 months. Three-month observation periods were used most often to evaluate the long-term efficacy of the test products.

From Table 1, it can be seen that all the studies except for one showed a statistically significant clinical benefit of HAP in reducing DH. In those studies where HAP was applied professionally, immediate relief of DH was achieved. HAP helped in the reduction of post-bleaching sensitivity in DH. At home application of HAP was in the form of gels in custom trays, but mostly it was in the form of toothpaste used twice a day. One study found that adding HAP to chewing gum worked to reduce DH. Compared to the placebo, HAP reduced DH from 6% to 80%. HAP was as good as or better than fluoride controls in reducing DH and as good or better than other desensitizing agents in reducing DH. This was confirmed in the meta-analyses of those studies that could be included in the meta-analyses (see Section 3.4 below).

#### 3.3.2. Risk of Bias

The Risk of Bias (RoB) assignments of 44 clinical trials included in the qualitative synthesis are shown in Table 2. Of those, 23 had low risk of bias, 15 had high risk of bias, and the remainder fell in between those ratings.

### 3.4. Quantitative Synthesis—Meta-Analysis

Three separate forest plots were generated from the meta-analysis. These included a comparison between HAP and placebo (Figure 2), between HAP and fluoride (Figure 3), and between HAP and other desensitizers (Figure 4). The results showed that HAP worked as well or better than fluoride or other desensitizing agents in reducing DH. The degree of mean relative reduction ranged from 10.2% (CI 95% [21.76; −19.26]) reduction in the HAP-group compared to positive controls, mean relative reduction of 23% (CI 95% [34.18; 11.82]) in the HAP-group compared to fluoride, and mean relative reduction of 39.5% (CI 95% [48.93; 30.06]) in the HAP-group compared to placebo. Figure 2, Figure 3 and Figure 4 show the forest plots of the main outcome comparisons.

## 4. Discussion

Two of the most common clinical problems of dentition for which patients seek professional help are dental decay and hypersensitive teeth. While the former can lead to tooth loss and the latter is more of an annoyance, making consuming foods and beverages of different temperatures and sweetness very uncomfortable, both conditions can benefit from the attention of a preventive dentistry professional before the problem becomes too difficult to manage. In our last systematic review, we focused on the anti-caries efficacy of HAP in toothpaste to reduce dental caries in children [18]. Here, we have turned our attention to the ability of HAP toothpaste to manage dentin hypersensitivity in adults, a very common condition. The prevalence of DH varies greatly, but based on a meta-analysis [6], at least every third adult, on average, suffers, or has suffered from the condition. 

HAP has been used for decades in toothpaste in Japan, where it was first developed, and in other countries (e.g., Germany), but it is a relatively new product in North America. Despite its widespread use in other dental applications, such as coating dental implants, bone repair, and periodontal surgery (see review by Chen et al. [68]), it has only recently become an accepted ingredient in oral care products in North America. Appendix A lists those products for use in Canada approved by Health Canada for sensitive teeth. Most products for use in the USA can be purchased through online importers, such as Amazon.com.

### 4.1. How HAP Reduces Dentin Hypersensitivity (DH)

Many in vitro experiments have shown that HAP in toothpaste and other oral care products adhere to the exposed dentin surface, coat the surfaces with microscopic particles of HAP, and occlude open dentin tubules, thereby reducing fluid flow and blocking pain signals from the odontoblast processes to the brain [69]. In addition, studies have shown that a lower plaque pH (or lower pH from dietary exposure) encourages dissociation of the HAP particles into calcium and phosphate ions [66,70,71]. Existing HAP crystals grow in the presence of excess calcium and phosphate ions. These are provided by the dissociation of HAP particles supplied by the oral care products exposed to a lower pH. The calcium and phosphate ions diffuse further into the dentin tubules. The pH is higher in the tubules the further from the surface the ions diffuse, and eventually salivary buffers neutralize the weak acids. This encourages the mineral phase to remineralize, grow, and occlude the tubules further than the physical obstructions provided by the HAP particles, which are still intact. When added to oral care products that are used regularly at home simply by brushing teeth twice a day, HAP-particles can serve to block dentin tubules and contribute to their remineralization (Figure 5).

The oral care products used in the studies listed in Table 1 varied in their synthetic HAP particle sizes. HAP is synthesized in various processes, which leads to a variety of particle dimensions. Most particles are micrometer in size, and can sometimes be measured in nanometer widths, as shown in many SEM studies in vitro. With dentin tubule width in the µm range (Figure 5), the small dimensions of the nano- and micro-HAP particles explain how they can accumulate in the dentin tubules and eventually occlude them, reducing dentin hypersensitivity. Specific crystal morphology and size have not been studied in detail to determine what the optimum dimensions for biomimetic HAP should be for dentin tubule occlusion and dentin adherence.

### 4.2. Strength of Evidence and Results of the Meta-Analysis

Our meta-analysis clearly demonstrates that there are many well-conducted clinical studies that show a significant lowering of DH in patients who report (or test positive for) hypersensitive teeth due mainly to exposed root surfaces after gingival recession. Overall, a significant reduction of 39.5% (CI 95% [48.93; 30.06]) can be expected when HAP toothpaste is used exclusively for a few weeks, compared to the placebo. Evidence from well-conducted clinical trials that HAP is an effective dentin desensitizer comes from many studies already published and reviewed (see Appendix A, which contains recent systematic reviews). We have updated the literature in the present systematic review and meta-analysis, having found 44 clinical trials on HAP in oral care products to desensitize sensitive teeth. Our meta-analysis of 22 RCTs is the most up-to-date quantitative synthesis of the evidence, indicating that HAP is an effective dentin desensitizer.

Enax et al. [72] reviewed the safety of calcium phosphates, including biomimetic HAP, and it was concluded that HAP can be safely swallowed when used in oral care products. However, most subjects expectorate and rinse after using their toothpaste. Toothpastes are not the only method of applying HAP to sensitive teeth. At home custom tray application can be a method of application. Leaving the HAP in contact with exposed dentin for longer periods of time may increase its efficacy, but more RCTs are required to determine if this is truly the case. The use of HAP products in patients after vital bleaching or periodontal therapy, whitening teeth with carbamide peroxides either professionally, or with home use products, can increase dentin hypersensitivity [73]. Even though the hypersensitivity is transient and thought to be the result of inflammation of the pulp, designers of in-office vital bleaching gels are testing whether the addition of HAP to their products can reduce after-treatment DH [74]. Those patients who have already experienced exposed dentin before bleaching could benefit from using HAP gels and toothpastes.

There were 11 clinical trials in which the test HAP-containing product was applied professionally by the clinician. In some of those studies, the product applied (a toothpaste) was also used at home. Apart from one study [20], the trials involving professional application were not included in the meta-analysis because the data from these studies were not comparable. However, all studies where HAP was professionally applied showed a benefit of using HAP-products as professional (in-office) treatment with respect to reducing DH [23,24,25,31,35,39,47,49,60]. Patients with exposed dentin may have very uncomfortable hypersensitivity after vigorous periodontal therapy (root planing) or after periodontal surgery [75]. The studies we found showed that patients can be helped after their periodontal surgery to manage dentin hypersensitivity when their root surfaces have been exposed.

### 4.3. Enhancing HAP Efficiency

There is some evidence that adding other elements to HAP might improve its ability to occlude dentin tubules and provide more stability to deposited crystals. Examples include Zn, Mg, and both [76]. Fluoride is thought to promote the remineralization potential of tubules, and fortifying fluoride toothpastes with HAP to improve the desensitizing potential of fluoride is a strategy that has not been fully tested. Novamin (calcium sodium phosphosilicate) added to fluoride toothpaste seems to be an effective strategy [77], but the evidence suggests that HAP outperforms other methods of desensitization [13].

### 4.4. Additional Studies

After our search was completed, 4 additional studies appeared in the literature that were not included in this qualitative or quantitative synthesis [78,79,80,81]. These studies were consistent with HAP in reducing dentin hypersensitivity. In all, 48 trials have been published examining the effectiveness of HAP as a dentin-desensitizing additive in oral care products.

### 4.5. HAP Toothpaste Approved for Use in Canada

Government regulatory agencies have strict regulations for making claims on toothpaste packaging. Without clinical evidence, claims of dentin desensitizing cannot be made. In Canada, a number of toothpastes have received permission from Health Canada to be sold with claims to treat sensitive teeth. See Appendix A in the Appendix A for this list and their characteristics.

## 5. Conclusions

Based on this systematic review and up-to-date meta-analysis, it can be concluded that hydroxyapatite is a safe biomimetic ingredient in oral care products for the reduction of dentin hypersensitivity, in addition to its already demonstrated anti-caries effects. Dental professionals can consider recommending hydroxyapatite-based oral care products as a primary strategy for the effective management of dentin hypersensitivity, which provides their patients with immediate and long-lasting relief from the dental pain caused by dentin hypersensitivity.

## Figures and Tables

**Figure 1 biomimetics-08-00023-f001:**
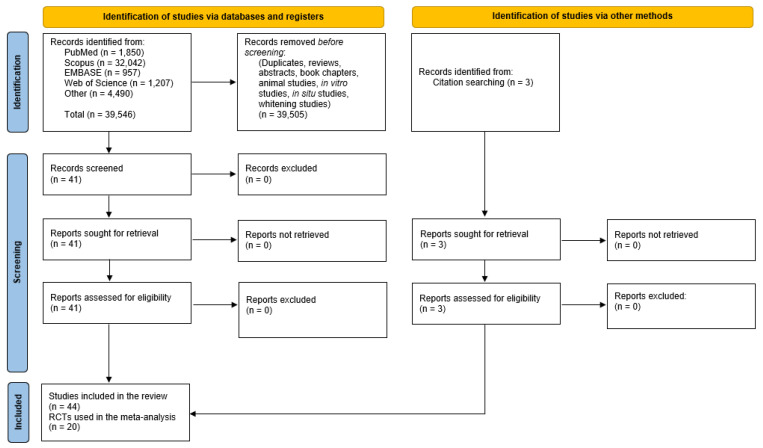
Flow diagram summary of the systematic review search strategy and results.

**Figure 2 biomimetics-08-00023-f002:**
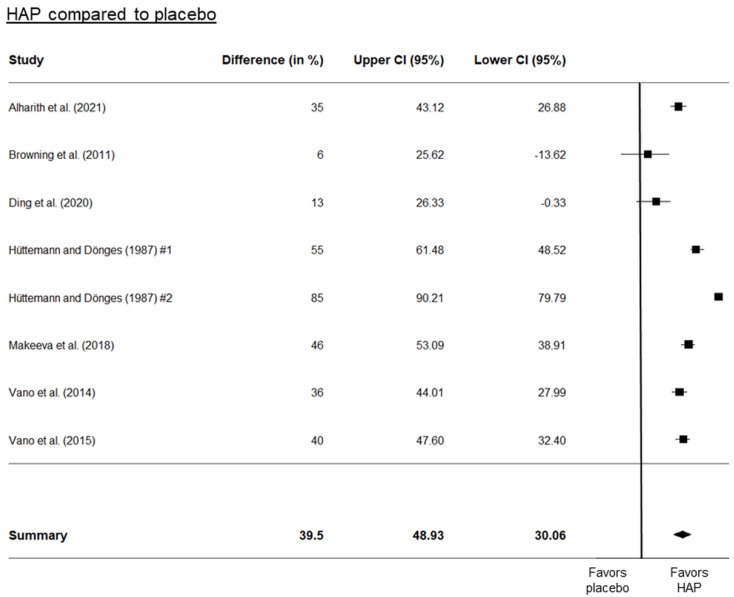
Forest plot of HAP compared to placebo. Hüttemann and Dönges [40] tested the different particle sizes of HAP. #1 indicates a particle diameter of 6 µm, and #2 indicates a particle diameter of 2 µm. Details of the studies in this figure are in Table 1. Other references: Alharith [24], Browning [32], Ding [36], Makeeva [51], Vano [61,62].

**Figure 3 biomimetics-08-00023-f003:**
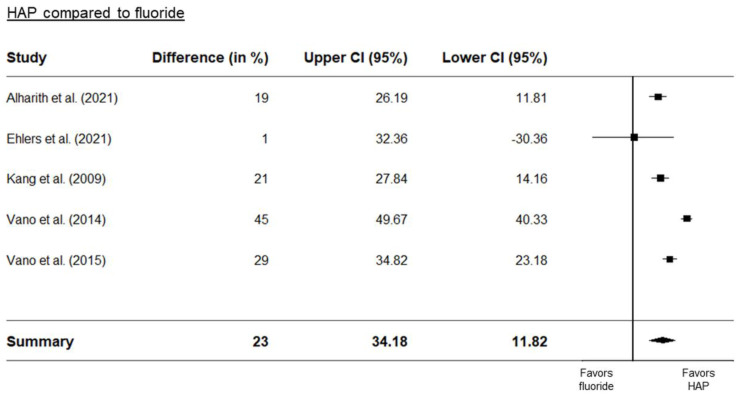
Forest plot of HAP compared to fluoride. Details of the studies in this figure can be found in Table 1. References: Alharith [24], Ehlers [37], Kang [42]. Vano [61,62].

**Figure 4 biomimetics-08-00023-f004:**
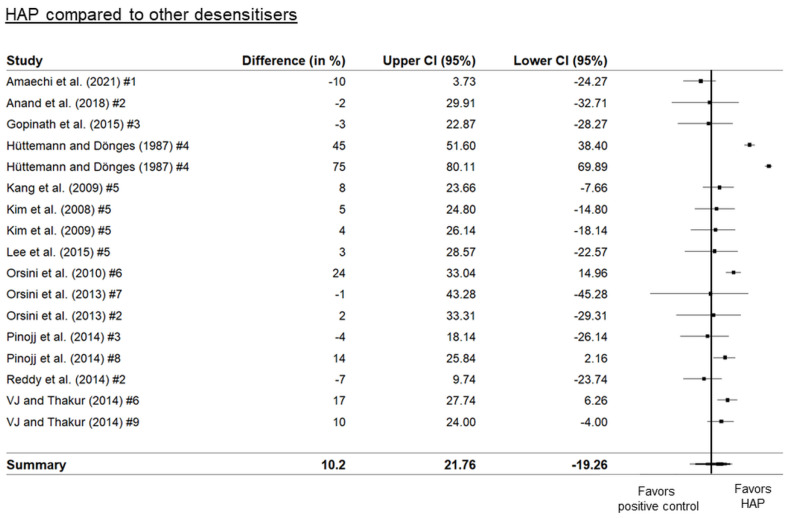
Forest plot of HAP compared to other desensitizing agents. Numbers indicate different types of agents: #1: 20% silica, #2: arginine, #3: Bioglass (calcium sodium phosphosilicate), #4: 0.125% benzocaine, #5: SrCl_2_ combined with CaCO_3_, #6: KNO_3_, #7: strontium acetate, #8: casein-phosphoprotein amorphous calcium phosphate (CPP-ACP), #9: propolis. Details of the studies in this figure and the ingredient comparisons are provided in Table 1. References: Amaechi [27], Anand [29], Gopinath [38], Hüttemann [40], Kim [43,44], Lee [46], Orsini [52,53], Pinojj [55], Reddy [58], VJ and Thakur [64].

**Figure 5 biomimetics-08-00023-f005:**
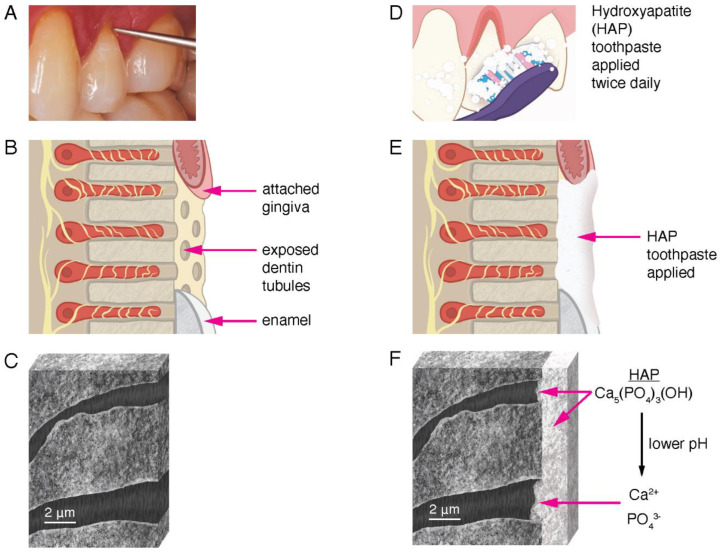
Illustration of the mechanism of HAP reduction of dentin sensitivity. (**A**): Clinical photograph of gingival recession on tooth 24, showing exposed dentin at the cervical margin indicated by the tip of the explorer. Tactile sensitivity occurs when the explorer makes contact with the exposed dentin. (**B**): Drawing of the area in (**A**) showing exposed dentin tubules. The attached gingiva, exposed dentin tubules, and enamel are labeled (not to scale). The yellow structures are nerve endings extending into the dentin tubules in close proximity to the odontoblast processes. The movement of liquid through the tubule elicits dentin hypersensitivity. (**C**): Drawing of an actual SEM image at high magnification of a section of exposed and demineralized human dentin showing two open dentin tubules. (**D**): Drawing of toothbrush application of hydroxyapatite toothpaste to the area in (**A**). (**E**): Drawing of the same area in (**B**) showing a layer of HAP toothpaste after one application. (**F**): After multiple uses, the entire dentin surface (in the before illustration in (**C**)) is covered with a layer of HAP. The dentin tubules are also occluded with HAP minerals. Both intact HAP molecules and ions from the dissociation of HAP contribute to the remineralization of the dentin surface, reducing dentin sensitivity. The HAP layer is stable with continued use of HAP toothpaste, producing a long-term reduction in dentin hypersensitivity.

**Table 1 biomimetics-08-00023-t001:** Summary of all clinical trials of hydroxyapatite (HAP) treatment of dentin sensitivity with GRADE assignments.

Study Author (Country)	Subjects	HAPProduct	Controls	Study Design and Length	Experi-mental Conditions	BlindingandRandom-ization	*p*-Value(</=)	Exam-iner Cali-bration	Study Conclusion	Comments	Quality of Evidence	GRADE Graphic
Al Asmari & Khan, 2019(Saudi Arabia)[22]	72 adults20–70 years	Biorepair(20% Zn-carbonate hydroxy-apatite) (Zn-CHA)	none	Clinical trial8 weeks of 2 times/day brushing with the toothpasteBaseline + 2 follow up exams	-air blast then Schiff sensitivity scale	Not Report-ed(NR)	0.001	Kappa = 0.83	“The use of the desensitizing toothpaste containing Zn-CHA in patients with DH provides significant rapid relief from DH.”	-a before and after trial design with no control-no blinding or randomiza-tion	LOW	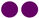
Alencar et al., 2020(Brazil)[23]	32 adult subjects	1. nHAP tooth-paste (simul-ated light)2. nHAP tooth-paste after laser light treatment	3. laser light + HAP-free toothpaste4. simulated laser light + nHAP-free toothpaste	1-month clinical trialbaseline, 1, 2 applications, then 1 month measure-ment-toothpastes used at home between treatments	Tactile and air blastVAS scale	Double blinded and random-ized	0.05	NR	“Intragroup analysis showed that only the GnHAP (simul-ated light and nHAP toothpaste) group showed a regression of DH at 1 month for the two applied stimuli.”	Small RCT with significant reduction in the nHAP toothpaste group	MODER-ATE	
Alharith et al., 2021(Saudi Arabia)[24]	63 adults18–60 years (mean age 39 years)	1. Nano-XIM (15% HAP)	2. Fluorophat Pro (5% NaF)3. Glycerin water placebo	1 week RCT	-explorer tactile stimulusor-cold air blast-then Schiff sensitivity scale	Double blinded	0.001	Kappa = 0.76, 0.79	“Within the limitations of the study, n-HA paste was the most effective desensitizing paste compared to fluoride and placebo pastes.”	-a well conducted RCT to test relief of dentin sensitivity after 1 week use of the test paste	HIGH	
Alsen et al., 2022(Brazil)[25]	30 adult subjects20 to 50 years	1. Nano-P containing nHAP (+9000 ppm fluoride, 5% KO_3_)	2. Flor-opal (0.5% fluoride, 3% KNO_3_)3. H_2_O	1-month RCTsingle in office application before vital bleaching	Air stimulation followed by Numerical Rating Scale (=VAS) scores	Blinding not possible for the exam-iner-patient partially blinded	0.05	NR	“Nanohydroxyapatite was more effective than fluoride, the commonly used material in this field, in reducing DH instantly after its application, though both materials had similar effects two- weeks and one-month post application.”	-in office one-time application-low subject numbers	LOW	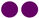
Amaechi et al., 2018(USA)[26]	52 adult subjects18 to 80 years	Apadent Pro (20% HAP) dental cream	20% silica cream	-ribbon of cream applied in tray for 5 min after brushing each evening before bed 8-week RCT with evaluations at 2, 4, 6, and 8 weeks	4-point Dental Pain Scale + VAS scale after cold and air stimulation	Double blinded, random-ized	0.001	Kappa Values = 0.80/0.88 (Air VAS), 0.87/0.89 (Air DPS), 0.91/0.94 (Cold VAS), 0.90/0.89 (Cold DPS)	“Within the limits of this study, it can be concluded that 20% nHAP dental cream is an effective method to promote the relief of DHS symptoms when applied daily.”	-a well conducted RCT	HIGH	
Amaechi et al., 2021(USA)[27]	105 adult subjects18 to 80 years	1. 10% n-HAP paste2. 15% n-HAP paste	3. 10% n-HAP + 5% KNO_3_4. CPSC + Na-MFP (1450 ppm fluoride)	2 times brushing per day8-week RCT with evaluations at 2, 4, 6, and 8 weeks	Endo-ice cold testair stimulation	Double blinded, random-ized	0.001	Kappa Values = 0.91/0.94 (Cold VAS) 0.80/0.88 (Air VAS)	“… it can be concluded that toothpaste containing nano-HAP alone (10 or 15% nano-HAP) or supplemented with KNO_3_ (10%nano-HAKN) was effective in relieving DHS symptoms when used at least twice daily. The study further demonstrated that the toothpaste containing 15% nano-HAP was more effective in sensitivity reduction than that containing 10% nano-HAP.”	-a well conducted RCT-dose response demon-strated	HIGH	
Amin et al., 2015(India)[28]	30 adult subjects20 male, 10 female	Aclaim (15% Hap)	none	6-month trial-toothpaste used at home-evaluations at baseline, 1, 3, and 6 months	-air stimulation-ice waterthen VAS	NR	0.0001	NR	“This study proves the efficacy of nano- hydroxyapatite paste in treating dentinal hyper-sensitivity.”	-a before and after trial design with no control-no blinding or randomization	LOW	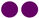
Anand et al., 2018(India)[29]	60 adult subjectsmean age 4242% males	1% nHAP toothpaste	Pro-Argin sensitivity fluoride toothpaste	4-week trial-toothpastes used at home-evaluations at baseline, 5 min, 1 week, and 4 weeks	-tactile test, and air stimulation then VAS for pain-digital electric pulp tester recordings	Double blinded-randomization by comp-uter and allot-ment carried out by another clinician	0.000	NR	“It appears from this study that both nHA based and arginine-based toothpastes are useful in the management of dentin hypersen-sitivity.”	-a well conducted RCT	HIGH	
Barrone & Malpassi, 1991(Italy)[30]	40 adult subjects	15% HAP paste	No control	6-month trial-toothpastes used at home-evaluations at baseline, 1, 2, 4, 12, and 24 weeks	Dental pulp test	NR		NR	“The topical application of a 15% gel of supermicron hydroxylapalite dentin according to our clinical experiences leads to an almost complete resolution of the symptoms in a very short time.”	A longitudinal before and after study showing effective reduction in dentin sensitivity (no control)	LOW	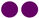
Bevilacqua et al., 2016(Brazil)[31]	30 adult subjects18 to 60 years of age	Desensi-bilize Nano-P (contain-ing 20% nHAP + 9000 ppm fluoride, 5% KO_3_)	1.23% fluoride gelBiosilicate	3-month split mouth design, professional application of either fluoride or biosilcate followed by Nano-P at baseline, week 1, 2 and 3 -final analysis at 3 months	Air blast stimulus10-point VAS scale	Double blindedRandomization method not reported	0.05	NR	“It can be concluded that there were no significant differences between treatments evaluated and, at the end of three months, all tested desensitizing agents reduced dentin hypersen-sitivity.”	A 3-month RCT with professional application-HAP was suspected of helping to reduce DS but did not show clearly because of the RCT design	LOW	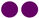
Browning et al., 2011(USA)[32]	42 adult subjects	Renamel After-Bleach)nHAP-paste	Zero nHAP placebo	2-week trial-desensitizer paste applied twice daily by tray 30 min after at home 7% hydrogen peroxide tray bleaching-Aim fluoride toothpaste without desensitizer was used by all subjects	-diary-based VAS pain scores recorded by subjects daily for 2 weeks	-double blinded -alloca-tion by random-ization chart	0.001	NR	“Within the limits of the study it can be concluded that use of a nano-hydroxyapatite paste following application of a tooth whitening agent was associated with a statistically significant reduction in the duration of tooth sensitivity.”	-no supervised pain stimuli used-diary method of subjective VAS pain score-method-ologically weak design	LOW	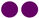
Choi et al., 2014(Korea)[33]	24 adult subjects21 to 61 years of age-average age 38.1 years	10% HAP + 19% TCP in commer-cial fluoride tooth-paste	Fluoride toothpaste control	4-week clinical trialbaseline, 1, 2, and 4 weeks	Cold water test10 cm VAS scale + 4-point Verbal rating scale	NR	NR	NR	“The toothpaste made with hydroxyapatite and tricalcium phosphate significantly relieves pain depending on the period of use.”	Statistical reduction in HS when HAP and TCP are added to fluoridated toothpaste-unable to show if it was the HAP or TCP	LOW	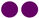
Da Silva et al., 2018(Brazil)[34]	60 adult subjects18 to 26 yrs of age	1. Nano-P containing 20% nHAP (+9000 ppm fluoride, 5% KO_3_)	2. Colgate Sensitive ProArgin3. Contene Organic without desensitizing additives	3-month trial5 min application after bleaching then 3 times/day brushing with the paste	Sensitivity scores from 0 to 4-eval-uations before bleaching, 1- and 10-days during bleaching, then at 1, 2, and 3 months	Randomization by numerical draw	0.05	NR	“The subjects treated with n-HAP and arginine presented lower sensitivity when compared to the control group.”	Well done RCT with ordinal scores	HIGH	
de Oliveira et al., 2016(Brazil)[35]	8 adult subjects, 138 sensitive teethage 24–46years	1. Nano-P contain-ing 20% nHAP (+9000 ppm fluoride, 5% KO_3_)	2. Sensodyne Rapid Relief3. Colgate Sensitive Pro-Relief4. Fluoride-free toothpaste (Cocorico)	10 s (Nano-P) to 60 s applications (digital vs. hand-piece brush)-in-office single application	VAS score immediately, at 1 day and at 30 days	double blindedsealed random allocation by an independent researcher	0.001	Calibrated examiners with an interclass corre-lation of 0.99	“The only desensitizing toothpaste that provided an immediate relief effect after both stimuli was that composed of calcium phosphate nanoparticles in the form of hydroxy-apatite.”	-in-office one time applicationlow subject numbers but adequate number of teeth compared	MODER-ATE	
Ding et al., 2020(China)[36]	45 adult subjects18 to 60 years	20% n-CAP (nanocarbonate apatite)(Denti-guard Sensitive)	Placebo toothpaste	6-week RCT-toothpaste applied 2 times/day at home	Air blast then VAS and Schiff Cold Air sensitivity scores12 to 24 h after root planning, then at 0, 2, 4, and 6 weeks	Double blindedRandom-ization by com-puter	0.001	NR	“The application of n-CAP-based dentifrice after non-surgical periodontal therapy could had some benefit on the reduction of DH after 4-week at- home use compared to the control dentifrice.”	Well done clinical trial showing the test toothpaste was signifi-cantly better than placebo in lowering dentin sensitivity at 6 weeks	HIGH	
Ehlers et al., 2021(Germany)[37]	21 subjects with MIHage 6 to 16 years	10% HAP (Kinder Karex)	1400 ppm amine fluoride toothpaste(Elmex Junior)	2-month (mean = 56 days) trial-toothpaste applied 2 times/day	Air blast then Schiff Cold Air Sensitivity Score+Tactile stimulus followed by Wong–Baker FACES pain rating scale	Double blindedSAS comp-uter gener-ated random-ization with age stratifi-cation	0.013	NR	“Both toothpastes (hydroxy-apatite versus amine fluoride) were effective in re- lieving hypersen-sitivity and maintaining desensitisation for 8 weeks.”	Well done clinical trial showing HAP works as well as amine fluoride to reduce sensitivity in young patients with MIH	HIGH	
Gopinath et al., 2015(India)[38]	36 adult subjectsage 18 to 60 years	n-HAP (Acclaim)	5% calcium sodium phosphor-silicate (CSP)(Shy-NM with NovaMin)	4-week clinical trialtoothpaste applied 2 times/day at home-baseline and 4th week measure-ments	tactile, air, then cold water applica-tions (in order, 5 min apart) then 10-point VAS scale scores	Double blindedRandomization allocation not reported	0.000 to 0.004	NR	“NovaMin and nano- HAP showed significant reductions in dentine hypersensitivity at the end of 4 weeks.”	A double blinded trial with just one time point measured after toothpaste use	MODER-ATE	
Gümüstas et al., 2021(Turkey)[39]	64 subjects18 to 40 years of age	30% n-HAP in alcohol (Prof. Oral Care nHAP Desenstizer)	CPP-ACP (Tooth Mousse)2.09% NaF (Ionite)placebo	1 week trial after vital bleaching-application was made for 4 min prior to bleaching	Air blast stimulationFollowed by 5-point VAS scale scores	Triple blinded (patient, operator and evalu-ator all masked to group assign-ment)-third person did random-ization-method not reported	0.05	NR	“Remineralization agents used for the treatment of post-operative sensitivity from tooth bleaching reduces the severity of the hyper-sensitivity, but does not prevent it from happening”	An in-office, single application trial with 1 and 7 day follow up	MODER-ATE	
Hütte-mann & Dönges, 1987(Germany)[40]	140 adult subjects20 to 60 years old	A: 17% HAP (6 µm)H: 17% HAP (2 µm)	B: 17% saltC: 0.125% benzocaineD: placeboE: 9% HAP. 8% salt, 0.125% benzocaineF: 17% HAP, 6% SrCl2G: 17% HAP, 5% SrCl2, 1% amine fluoride	1 to 2-week trial-paste applied at home	Standard-ized cold test-question-naire results	NR	NR	NR	“The efficacy of finely granular hydroxyapatite in the treatment of dentine sensitivity was demonstrated, 90% of the subjects indicated improvement after 3 to 5 days, 50% were pain-free within the period of the study.”	Trial comparing multiple pastes-subjective patient reporting of home experience with pastes	LOW	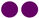
Jena & Shash-irekha, 2015(India)[41]	45 adult subjects age 18 to 50 years	15% HAP (nano-XIM)	5% NovaMin (Vantej)8% arginine Colgate Sensitive Pro-Relief)	4-week trialone time application of paste	-Tactile test10-point VAS scale-Air blast -Schiff Cold Air Sensi-tivity Score-evalua-tions at 1 and 4 weeks	Double blindedRandom-ization with comp-uter-gener-ated table	0.05	NR	“15% n-HA containing toothpaste was found to be most effective followed by 8% arginine and 5% NovaMin group.”	RCT with one time application	HIGH	
Kang et al., 2009(Korea)[42]	150 adult subjectsmean age of 35 years	HAP tooth-paste (Diome Plus PRTC, Korea)	Fluoride toothpaste (2080 Korea)Strontium chloride toothpaste (Senso-dyne GSK)	4-week trialassessments at 1 and 4 weeks-toothpastes used at home	Ice test stimulus-11-point VAS	NR	0.0001	NR	“Toothpaste containing hydroxyapatite is effective in reducing hyper-sensitivity.”	Clinical trial showing HAP toothpaste reduced DS	LOW	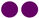
Kim et al., 2008(Korea)[43]	100 adult subjectsage 22 to 69 yearsmean age 47.2 years	10% HAP(Diomi-plus PRTC, Korea)	Strontium chloride toothpaste (Senso-dyne, GSK)	4-week trialassessments at 1 and 4 weeks-toothpastes used at home	cold test stimulus11-point VAS	NR	0.0001	NR	“The toothpaste containing apatite showed statistically significant similar results to the toothpaste containing strontium chloride, which is known to have a significant effect on hypersensiti-vity through several previous studies.”	Clinical trial showing HAP toothpaste reduced DS as well as strontium chloride toothpaste	LOW	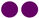
Kim et al., 2009(Korea)[44]	55 adult subjectsmean age of 43.5 years	10% HAP(Diomi-plus PRTC, Korea)	Strontium chloride toothpaste (Sensodyne GSK)	8-week trial-baseline, 2-, 4- and 8-week assessments-toothpastes used at home	Cold water and air blast stimulus10 cm VAS scale + Verbal rating score	Double blinded (method not re-ported) Random-ized (method not re-ported)	0.05	NR	“The toothpaste containing hydroxyapatite showed no statistical difference in reducing hyper-sensitivity from the toothpaste containing strontium chloride, which was previously known to be effective for dental hyper-sensitivity, and showed significant clinical improvement during the 8-week period of use”	RCT showing HAP toothpaste reduced DS as well as strontium chloride toothpaste	MODER-ATE	
Kondyurova et al., 2019(Russia)[45]	60 adult subjectsage 18 to 65(mean = 39.5)	0.5% nHAP (SPLAT Sensitive Ultra)	0.1% nHAP (Splat Profess-ional Sensitive White)	4-week trial -exam-inations at baseline, 2 and 4 weeks	air blast (using Schiff sensitivity score) after tactile then chemical stimuli, both scored with a 4-point scale	Tooth-pastes handed out in original package labelsCom-puter-gener-ated random-ization	0.05	NR	“In conclusion, the results of this study support the short term efficacy of a x% nHAP occlusion technology-based toothpaste for the relief of dentin hyper-sensitivity.”	Not blinded-significant reduction of dentin sensitivity for both concen-trations of HAP	LOW	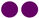
Lee et al., 2015(Korea)[46]	82 adult subjects20 to 65 years of agemean age 37.2 years	1. 20% n-CAP (Carbon-ated HAP), 8% silica (Denti-guard Sensitive)	2. 10% CaCO_3_, 10% SrCl2 (Senso-dyne, GSK)3. Laser treatment	4-week trial -profession-al laser treatment at baseline, week 1 and week 2-test toothpastes used at home 2 times/day-then standard fluoride toothpaste used for the remaining 2 weeks	Tactile and air blast sensitivity10 cm VAS scale + 4 point Schiff scores	Single blindedRandom-ization claimed but not reported	0.05	Two exam-iners were cali-brated Kappa not reported	“The use of both the desensitizing dentifrices containing 20% n-CAP as self-care and the Er,Cr:YSGG laser as professional treatment were effective in reducing dentin hyper-sensitivity.”	A single blinded RCT showing carbonated HAP in silica reduced DS as well as laser and strontium chloride toothpaste	MODER-ATE	
Loguercio et al., 2015(Brazil)[47]	40 adult subjects22 to 24 years of age	Nano-P (20% nHAP + 9000 ppm fluoride + 5% KO_3_)	Placebo paste	2-day trial-paste was applied in office before vital bleaching	Tooth sensitivity was recorded using a numeric rating scale (0–4) during bleaching and up to 48 h after each session.	Double blinded-the pack-aging was the same, but the placebo had a different consist-encyComp-uter-gener-ated random-ization tables	0.53	85% kappa agree-ment for patient allo-cation	“The use of a nano-calcium phosphate paste containing potassium nitrate, fluoride, and calcium phosphate prior to in-office bleaching did not reduce bleaching-induced tooth sensitivity measured during and up to 48 h after each session.”	Short trial-no significant de-sensitizing-placebo was missing fluoride and KNO_3_ so no conclusion could have been reached about HAP	LOW	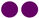
Low et al., 2015(USA)[48]	60 adult patients18 to 75 years of age47 females, 13 males	AO Pro Tooth-paste sensitive(KNO_3_, MFP, nHAP, phloretin, ferulic acid, syli-marin)	No control	2-week trial-assessment at baseline and then at home after 2 weeks using the toothpaste	Question-naire -five questions, rated on a 10-point scale, asking (i) degree of pain, (ii) duration of pain, (iii) intensity of pain, (iv) tolerability of pain, and (v) description of pain.	NR	0.001	NR	“The outstanding results of speed and effectiveness of the commercially available toothpaste suggest the contributing activity of the newer nano-hydroxyapatite and of the polyphenol antioxidants.’	Not able to determine which ingredient worked-results qualitative-no control	VERY LOW	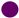
Maharani et al., 2012(Indon-esia)[49]	20 adult subjects	A: commer-cial HAP tooth-paste with potas-sium and zinc C: citrates, MFP and phos-phate	B: placebo (no active ingre-dients)	8-hr trial measuring DS at baseline, 30 sec after application of the paste and after 8 hr	Electric pulp tester followed by a 10-point VAS	Double blinded	0.05	Kappa = 0.87(intra-exam-iner)	“It may be concluded that treatment with (hydroxy-apatite toothpaste) effectively reduced dentin hyper-sensitivity. The effect was instant and long lasting.”	Short duration, low number of subjects clinical trial-cannot determine which ingredient was effective	VERY LOW	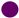
Makeeva et al., 2016(Russia)[50]	30 adult subjectsone group younger than the other17 to 44 years of age	Medical HAP(Apadent Total Care)	No control	3-month clinical trial-toothpaste used at home	Air blast sensitivity test4 point Schiff score	NR	NR	NR	“Long-term use of Apadent Total Care toothpaste effectively reduces tooth sensitivity in patients of different age groups.”	Simple trial –before and after design with no control	LOW	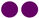
Makeeva et al., 2018(Russia)[51]	40 young adult subjects20–25 years of age	6% Nano-HAP (Innova paste)+1% Nano-HAP liquid (Liquid Enamel)	No paste, liquid control	14-day trialassessment at 3, 7, and 14 days	Air blast sensitivity test4 point Schiff score	NR	NR	NR	“Nano-HAP 6% paste and 1% suspensions can be used as an alternative replacement therapy in the treatment of enamel hyperesthesia.”	Simple trial –before and after design with no control	LOW	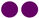
Orsini et al., 2010(Italy)[52]	75 adult subjects -between 18 and 75 years	30% Zn Carbon-ate Hydroxy-apatite tooth-paste(BioRe-pairPlus)	Sensodyne Pronamel	8-week trial-subjects brushedteeth 2 times per day for at least 1 min for the entire trial	Tactile, air and cold tests3-point sensitivity scale plus 10-point subjective pain scale	Double blinded Comp-uter-gener-ated random-ization	0.001 to 0.009	NR	“This trial represents the first clinical demonstration that nanostructured CHA microparticles may significantly reduce painful stimuli and could therefore be used as active ingre- dients for desensitizing dentifrices.”	Well done clinical trial with significant reduction in dentin sensitivity	HIGH	
Orsini et al., 2013(Italy)[53]	90 adult subjects18 to 75 years of age29 males, 69 females	30% Zn Carbon-ate Hydroxy-apatite tooth-paste(BioRe-pairPlus)	8% Arginine carbonate 1450 ppm F MFP (Colgate Sensitive)8% Sr Acetate, 1044 F NaF (Senso-dyne Rapid Relief)	3-day trialsubjects brushed teeth 2 times/day for the 3-day trial	Tactile, air and cold tests3-point sensitivity scale plus 10-point subjective pain scale	Double blinded Com-puter-gener-ated random-ization	0.003	One exam-iner, no Kappa statistics reported	“The three tested dentifrices significantly reduced DH after 3-day treatment, supporting their use in clinical practice.”	A short well done clinical trial	HIGH	
Park et al., 2005(Korea)[54]	44 adult subjects26 to 71 years of age	Tooth-paste with microcrystalline HAP	No control	8-week clinical trial-assess-ments at 2, 4, and 8 weeks	-cold, air, tactile stimula-tions10 cm VAS scale + 4-point verbal rating scale	NR	0.05	NR	“The toothpaste containing micro-crystalline hydroxyapatite has a relieving effect on various stimuli that cause hyper-sensitivity symptoms such as cold stimulation, compressed air stimulation, and tactile stimulation during the period of use for 8 weeks”	Simple trial –before and after design with no control	LOW	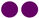
Pinojj et al., 2014(India)[55]	80 teeth per groupSubjects were 18 to 50 years of age	nHAP tooth-paste (SHY NM)	Calcium Phospho-silicate (SHY)CPP-ACP paste	3-month trial-baseline, 2, 4 weeks, 2, 3-month exam-inations after using toothpastes 2 times/day at home	Air and cold-water stimulus10-point VAS scale + Schiff base 3-point scores	Triple blindedRandom-ization method not reported	0.000	NR	“The nanoparticle hydroxyapatite group was found to be significantly better in reducing the visual analog scale score as well as Schiff test score and at any time point for both measures of sensitivity.”	A clinical trial showing superiority of nHAP toothpaste in lowering dentin sensitivity.	MODER-ATE	
Polyakova et al., 2022(Russia)[56]	30 adult subjects35–45 years of age	20% n-HAP paste	nZnMg-HAP positive controln-FAP positive control	1 month trialevaluated at baseline, 2 and 4 weeks	Air blast stimulus4-point Schiff sensitivity score	Double blindedRan-Domized(details not pro-vided)	0.00083	NR	“The nZnMgHAP-containing toothpaste provided a significant reduction in airblast sensitivity after 2 weeks of daily use in adult patients with cervical non-carious defects. This effect was significantly greater compared to pure nHAP and nFAP.”	The 20% n-HAP paste significantly reduced DH at 4 weeks compared to baseline.Mg and Zn seemed to improve the desensitisation effect	HIGH	
Porciani et al., 2014(Italy)[57]	100 adult subjects18 to 65 years of age	12 mg CaHAP + 97 mg diCaPhos-phate Dihy-drate (DPD) per 1.4 gm chewing gum stick	Placebo control	2-week trial-two chewing sticks 3 times/day for 2 weeks-exam-ination at baseline, 1 and 2 weeks	Air blast, tactile, cold-water test 3-score sensitivity test + 10-point subjective score	Double blindedRandom-ization method not reported	0.001 to 0.05	NR	“The group using the chewing gum containing calcium hydroxyapatite had a statistically significant reduction in all clinical test indexes for dentin hyper-sensitivity after one and two weeks, and a statistically significant reduction compared to the control gum group.”	A 2 week chewing gum blinded trial showed either CaHAP or DPD or both effectively lowered dentin sensitivity	LOW	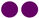
Reddy et al., 2014(India)[58]	30 adult subjects	Acclaim (15% HAP)	Colgate Pro-Argin	3-day clinical trial-toothpaste used at home	Air blast and cold-water stimu-lation10-point verbal rating scale	Blinding not reportedRandom-izationdetails not pro-vided	0.001	NR	“’Both the experimental dentifrices Pro-Argin and Acclaim were found to provide rapid relief in patients.”	Short trial, lower quality with significant results for both toothpastes in lowering dentin sensitivity	LOW	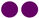
Seong et al., 2021(England)[59]	82 adult subjectsage 18 to 65 years	KNO_3_/Al/HAP/MFP(Sunstar)	5% KNO_3_/NaF (Senso-dyne)	2-week clinical trialbaseline, 1- and 2-week exam-inationstoothpaste used 2 times/day at home	Tactile then cold test3-point VAS scale + quality of life question-naire	Exam-iner blindingRan-dom-ization method not reported	0.001	NR	“This study demonstrated the efficacy of an aluminium lactate/potassium nitrate/hydroxyl-apatite toothpaste compared to a potassium nitrate control toothpaste for the prevention of dentine hyper-sensitivity both immediately and over a 2-week period.”	Single blinded RCT -HAP may have reduced DS, but positive control selection meant other ingredients (Al, MFP) could have helped	LOW	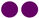
Shetty et al., 2010(India)[60]	486 teeth in 45 adult subjects-aged 28 to 42 years	A: HAP in dry sol powder B: HAP liquid	C: placeboD: notreatment	8-week trialIn-office application-baseline, + 1-, 2-, 4- and 8-week exam-inations	Tactile, cold water, air blast stimuli-Linear VAS scores, and verbal rated scores of 0 to 3	Single blinded-method of random-ization not reported	0.001	NR	“HAP shows definite potential as an effective desensitizing agent providing quick relief from symptoms.”	Single blinded, but well controlled clearly showing in office treatment with HAP reduces DS for 4 weeks but not 8 weeks.	LOW	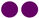
Vano et al., 2014(Italy)[61]	105 adult subjectsmean age 43 years	Group 1: 15% nHAP(Prev-Dent)	Group 2: Positive fluoride paste control (1500 ppm F as MFP- Colgate Cavity protection)Group 3: Prev-Dent Placebo	4-week RCT-baseline + 2- and 4-week exam-inations-toothpaste used 2 times/day at home	Tactile and air blast stimuli-VAS scale and examiner-based Schiff assessment	Double blindedRandom number gener-ator used for random-ization	0.001	Kappa statistics was con-ducted on inter-examiner perfor-mance on 10% of subjects but results not reported	“The findings of the present study encourage the application of nano-hydroxyapatite in fluoride-free toothpaste as an effective desensitizing agent providing quick relief from symptoms after 2 and 4 weeks.”	Well done HAP RCT showing HAP toothpaste desensitized dentin better than regular fluoride toothpaste and the placebo	HIGH	
Vano et al., 2015(Italy)[62]	60 subjects27 to 29 years of age	6% hydrogen peroxide with 2% nHAP (Prev-Dent)	6% hydrogen peroxide control group	A 2-week dentin sensitivity trial within a 9 mo. vital bleaching trial-the active ingredient was compared to the control at 1, 7, and 14 days	Tactile and air blast stimuli-VAS scale and examiner-based Schiff assessment	Random card gener-ated allo-cation-blinded to the exam-iner	0.05	Kappa Score of 0.89 on duplicate exam-inations of 10% of the subjects	“6% HP with 2% n-HA resulted in significant lower tooth sensitivity at 24 h post-treatment.”	Single blinded RCT with significant reduction of DS during vital bleaching	MODER-ATE	
Vano et al., 2018(Italy)[63]	105 adult subjectsaverage age of 39 yrs	2% nHAP tooth-paste (Cavex Bite and White ExSense)	Colgate Cavity Gel protectionPlacebo	4-week clinical trial-gel applied 10 min dailyexamination at baseline, 2 and 4 weeks	-cold air and tactile sensitivity100 mm VAS scale and Schiff base 4-point scores	Double blindedCom-puter-gener-ated random-ization table	0.05	Kappa statistics was con-ducted on inter-examiner perfor-mance on 10% of subjects but results not reported	“The application of nano-hydroxyapatite in gel toothpaste fluoride free is an effective desensitizing agent providing relief from symptoms after 2 and 4 weeks”.	Double blinded RCT showing significant reduction of DS by 2% nHAP gel	HIGH	
VJ Nar-mantha & Thakur, 2014(India)[64]	45 adult patients	1% nHAP(Acclaim)	5% KNO_3_ (Senso-dent-K)Propolis toothpaste	Air blast sensitivity	100 mm VAS scale	NR	0.03	NR	“It can be concluded that nanohydroxy-apatite and propolis are a potential treatment modality for dentin hyper-sensitivity.”	Small, not randomised and not blinded study	LOW	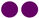
Wang et al., 2016(Brazil)[65]	28 adult subjects, 137 teeth-between 18 and 60 years old	Destab-ilize Nano-P (20% nHAP, KNO_3_, 9000 fluoride NaF)Home-care Nano-P (10% HAP, KNO_3_, 900 ppm F NaF)	Pro-Relief Pro-Argin (8% arginine)Duraphat varnish (26,300 ppm fluoride)	3-month clinical trial-varnish and professional paste applied profess-ionally at each appoint-ment but home care pastes applied at home-assessment at 1 and 3 months	Air blast stimulus10-point VAS scale	Double blindedRandomization by MS Excel program	0.94 (no diff-erence bet-weengroups)	Cali-brated exam-iners	“Nano-hydroxy-apatite formulations (with or without home-care product association) were as effective as the other treatments in reducing dentin hyper-sensitivity over three months.”	Mixed RCT with profess-ional + home care (patients likely not blinded to professional treatment)	MODER-ATE	

**Table 2 biomimetics-08-00023-t002:** Risk of Bias (RoB) assignments of the hydroxyapatite (HAP) and dentin hypersensitivity clinical trials.

Study	Tested Products	Application Protocol	Trial Length(Weeks)	RandomSequence Genera-tion ^a^(Selection Bias)	Allocation Conceal-ment ^b^(Selection Bias)	Blinding of Participants and Personnel ^b^ (Performance Bias)	Blinding of Outcome Data ^c^ (Attrition Bias)	Incomplete Outcome Data ^d^ (Attrition Bias)	Selective Reporting ^e^ (Reporting Bias)	Overall BIAS
Al Asmari & Khan, 2019[22]	Zn-carbonate HAPbefore and after trial-no placebo	At home toothpaste	8	NA	NA					
Alencar et al., 2020[23]	Nano-P HAP paste vs. placebo after laser	Profess-ional + at home toothpaste	4							
Alharith et al., 2021[24]	NanoXIM (15% HAP), Fluorophat Pro (5% NaF), placebo	Profess-ional(one time)	1							
Alsen et al., 2022[25]	Nano-P, Fluor-Opal, dH20 (placebo)	Profess-ional(one time)	4							
Amaechi et al., 2018[26]	one 5 min application each day in custom tray 20% nHAP cream vs. 20% Silica	At home (custom tray)	8							
Amaechi et al., 2021[27]	10%nHAP (+/− KNO_3_), 15% nHAP, CSPS	At home (toothpaste)	8							
Amin et al., 2015[28]	before and after Acclaim (1% HAP), no placebo or pos. control	At home (toothpaste)	24	NA	NA					
Anand at al, 2018[29]	Aclaim (1% HAP) vs. Colgate Sensitive Pro-Relief (8% Arginine)	At home (toothpaste)	4							
Barone & Malpassi, 1991[30]	before and after trial 15% HAP gel applied 10 sec. 3x/day	At home gel applica-tion 2 weeks	24	NA	NA					
Bevilacqua et al., 2016[31]	1.23% APF gel + nano-P vs. APF-gel + Biosilicate	Profess-ionalone time for 1 min	12							
Browning et al., 2011[32]	nHAP (Renamel AfterBleach)placebo control	At home (custom tray after bleaching)	4							
Choi et al., 2014[33]	10% HAP (+ F, TCP) vs. control (not specified)	At home toothpaste	4							
Da Silva et al., 2018[34]	20% HAP paste (Nano P) vs. placebo or Colgate Sensitive ProArgin (8% arginine)	At home toothpaste (after bleaching)	12							
De Oliveira et al., 2016[35]	20% HAP (Nano P) vs. Sesnodyne, Sensitive Pro-Relief, placebo	Profess-ional(one time)	4							
Ding et al., 2020[36]	20% HAP paste vs. placebo	At home toothpaste	6							
Ehrlers et al., 2021[37]	Kinder Karex (10% HAP) toothpaste vs. Elmex (amine fluoride, 1400 ppm fluoride)	At home toothpaste	8							
Gopinath et al., 2015[38]	Aclaim (1% HAP) vs. 5% CSPS paste	At home toothpaste	4							
Gümüştaş & Dikmen, 2021[39]	Oral Care Nano HAP (30% Hap) vs. placebo, Tooth Mousse, 2.09% NaF (Ionite)	Profess-ionalone time before bleaching	1							
Hüttemann & Dönges, 1987[40]	9 to 17% HAP(+/- benzocaine, SrCl_2_, Amine fluoride)	At home toothpaste	1+							
Jena & Shashire-kha, 2015[41]	15% HAP vs. 5% Novamin, 8% arginine toothpaste	Profess-ionalone time	4							
Kang et al., 2009[42]	HAP vs. SrCl_2_, fluoride toothpaste(compositions unknown)	At home toothpaste	4							
Kim et al., 2008[43]	Diomiplus PRTC (fluoride toothpaste with 10% HAP) vs. Sensodyne	At home toothpaste	4							
Kim et al., 2009[44]	Diomiplus PRTC (fluoride toothpaste with 10% HAP) vs. Sensodyne	At home toothpaste	8							
Kondyurova et al., 2019[45]	SPLAT 0.5% HAP vs. SPLAT (0.1% HAP)	At home toothpaste	4							
Lee et al., 2015[46]	Dentiguard Sensitive (20% carbonate HAP, 8% silica) vs. SrCl_2_, Laser	-Laser in office (twice) -at home toothpaste	4							
Loguercio et al., 2015[47]	Nano P vs. placebo	Profess-ional one time 10 min	2							
Low et al., 2015[48]	HAP toothpaste with MFP, KO_3_-no placebo control	At home toothpaste	2							
Maharani, 2012[49]	HAP toothpaste with zinc, TSP, MFP vs. placebo	Profess-ional one time	8 h							
Makeeva et al., 2016[50]	Apadent Total Care (7.0% HAP)	At home toothpaste	12							
Makeeva et al., 2018[51]	6% HAP paste + 1% INNOVA vs. no treatment control	At home toothpaste and rinse	2							
Orsini et al., 2010[52]	Biorepair (Zn-carbonated HAP) vs. ProNamel	At home toothpaste	8							
Orsini et al., 2013[53]	Zn-carbonate 30% HAP vs. 8% arginine +1450 ppm F, 8% SrAcetate + 1040 ppm fluoride toothpastes	At home toothpaste	3 days							
Park et al., 2005[54]	HAP toothpaste (content not provided), no control	At Home toothpaste	8							
Pinojj et al., 2014[55]	Aclaim (1.0 % HAP) vs. CSPS, CPP-ACP	At Home toothpaste	12							
Polyakova et al., 2022[56]	20% HAP paste vs. ZnMgHAP and F-HAP pastes	At home toothpaste	4							
Porciani et al., 2016[57]	HAP + dicalcium phosphate dihydrate chewing gum vs. palcebo	At home chewing gum 3 times/day	2							
Reddy et al., 2014[58]	Aclaim (1% HAP) vs. Coglate ProArgin	At home toothpaste	3							
Seong et al., 2021[59]	HAP (+KNO_3_, Al-lactate) vs. Sensodyne	At home toothpaste	2							
Shetty et al., 2010[60]	High % HAP slurry vs. placebo	Professional one time	8							
Vano et al., 2014[61]	15% HAP paste vs. fluoride paste vs. placebo	At home toothpaste	4							
Vano et al., 2015[62]	2% HAP in 6% carbamide peroxide bleach vs. CP bleach without HAP	At home gel applica-tion 2 weeks	2							
Vano et al., 2018[63]	Cavex (2% HAP paste) vs. Colgate Cavity Gel (1500 ppm fluoride in MFP) paste vs. glycerin placebo	At home gel applica-tion one time/day 10 min	4							
VJ Narmatha & Thakur, 2014[64]	Aclaim (1% HAP) vs. Sensodent-K (5%KNO_3_) vs. 10% propolis	At home toothpaste	4							
Wang et al., 2016[65]	Desensibilize Nano-P (20% HAP, 9000 ppm F, KNO3) vs. ProArgin, Prorelief, 5% NaF varnish	Professional applicationone time/week, 3 weeks	12							

**a Randomization: Was the allocation sequence random? Was the allocation sequence concealed until the participants were assigned to the intervention.** Did the baseline difference suggest a problem with the randomization process? **b Deviations from intended interventions:** Were participants aware of the assigned intervention? Were people delivering interventions aware of the participants’ assigned intervention? If yes, were there deviations from the intended intervention that arose because of trial context? If yes, were these deviations likely to have affected the outcome? If yes, were these deviations balanced between the groups? If yes, was an appropriate analysis used to estimate the effect of assignment to intervention? If no, was there potential for substantial impact on the result? **c Missing outcome data:** Were data for this outcome for all or nearly all participants randomized? If no, is there evidence that the result was not biased by missing outcome data? If no, could missingness of outcome data depend on its true value? **d Measurement of the outcome:** Was the method of measuring the outcome appropriate? Could measurement of the outcome have differed between intervention groups? If no, were outcome assessors aware of the intervention received by the study participants? If yes, could assessment of the outcome have been influenced by the knowledge of if the intervention received? If yes, is it likely that this occurred? **e Selection of the reported result:** Were the data that produced the results analyzed in accordance to the prespecified analysis plan that was finalized before unblinded outcome data were available for analysis? Is the numerical result being assessed likely to have been selected, on the basis of the results, from (a) multiple eligible outcome measurements? (b) multiple eligible analyses of the data?

## Data Availability

Not applicable.

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
