# Peer review of "Clinical Evidence of Biomimetic Hydroxyapatite in Oral Care Products for Reducing Dentin Hypersensitivity: An Updated Systematic Review and Meta-Analysis"

_biomimetics, 2023, doi:10.3390/biomimetics8010023_

Round 1
Reviewer 1 Report
Limeback et al. report a systematic review and analysis of the effect of hydroxyapatite in oral care products. They concluded that hydroxyapatite is a safe, biomimetic ingredient in oral care products to reduce dentin hypersensitivity and anti-caries effects. They suggest dental professionals recommend hydroxyapatite-based oral care products as a primary strategy for the effective management of dentin hypersensitivity. Moreover, the authors provide a discussion regarding its function in the reduction of dentin hypersensitivity. This may look more like a clinical report, but I suggest publication after a few minor revisions.
1. I am a little confused. This is a reviewer manuscript, and it appeared as a research article. Section 2 describes the materials and methods. Sections 3 and 4 introduce the results and discussion. In section 2, the authors introduced the way of searching. Generally speaking, the authors studied the clinical reports and summarized them. Finally, The conclusion is put forward. I am not sure whether this is a review or a research paper.
2. In the results, I would appreciate it if the authors could do a better summary of the clinical results.
3. Regarding the mechanism, the authors stated that “HAP in toothpaste and other oral care 230 products adhere to the exposed dentin surface, coat the surfaces with microscopic particles of HAP and occlude open dentin tubules” and “HAP-particles can serve to block dentin tubules and contribute to their remineralization”. This can lead to two questions: 1. Can any microparticles that can adsorb on teeth reduce dentin hypersensitivity? From the mechanism perspective, the authors’ analysis may lead to this conclusion. 2. It is well known that saliva is rich in proteins. How does protein desorption kinetics influence particle adhesion? Recent work (Lyu, Biomacromolecules, 2022, 4709) shows that protein desorption changes with time. The protein adsorption should influence the particle deposition, but how? Does it affect the usage of oral care products, usually twice daily, to reduce dentin hypersensitivity?
4. I would appreciate it if the authors could provide more discussions on the effect of pH on HAP deposition. Why and how does the dissociation of HAP contribute to the remineralization of the dentin surface?
Author Response
- I am a little confused. This is a reviewer manuscript, and it appeared as a research article. Section 2 describes the materials and methods. Sections 3 and 4 introduce the results and discussion. In section 2, the authors introduced the way of searching. Generally speaking, the authors studied the clinical reports and summarized them. Finally, The conclusion is put forward. I am not sure whether this is a review or a research paper.
Authors’ response:
The manuscript is structured as recommended in the Biomimetics instructions to authors, following the PRISMA guidelines for systematic reviews (reference 17 in the manuscript). According to Biomimetics, “Structured reviews and meta-analyses should use the same structure as research articles and ensure they conform to the PRISMA guidelines” (as presented here: https://www.mdpi.com/journal/biomimetics/instructions#preparation)
The following link: https://prisma-statement.org/Extensions/Abstracts shows the structure of reporting is similar to a research study.
- In the results, I would appreciate it if the authors could do a better summary of the clinical results.
Authors’ response:
This reviewer makes a good suggestion here. In the results section, the results reported in the manuscript include what techniques various researchers generally used to measure dentin hypersensitivity, their subjective scoring methods, and whether the studies were well conducted (Risk of Bias and GRADE scores). In the meta-analyses the results were divided into 3 main comparisons: HAP vs placebo, HAP vs fluoride and HAP vs other desensitizing agents. A discussion of those results was presented in the discussion section.
Direct quotes of the conclusion by each study author(s) and our comment about each study (summarizing the strengths and weaknesses) are provided in Table 1. But it would be useful to summarize in the text the overall impression of the clinical results (without providing discussions that are already in the discussion section). We have therefore added the following passage to the Results section at the end of Table 1.
"From Table 1 it can be seen that all the studies except for one showed a statistically significant clinical benefit of hydroxyapatite in reducing DH. In. those studies where HAP was applied professionally, immediate relief of DH was achieved. HAP helped in the reduction of post-bleaching sensitivity DH. At home application of HAP was in the form of gels in custom trays but mostly it was in the form of toothpaste used twice a day. One study found that adding HAP to chewing gum worked to reduced DH. Compared to placebo, HAP reduced DH from 6% to 80%. HAP was as good as or better than fluoride controls in reducing DH and as good or better than other desensitizing agents in reducing DH. This was confirmed in the meta-analyses of those studies that could be included in the meta-analyses (see 3.4 Quantitative synthesis-meta-analysis below)”.
- Regarding the mechanism, the authors stated that “HAP in toothpaste and other oral care 230 products adhere to the exposed dentin surface, coat the surfaces with microscopic particles of HAP and occlude open dentin tubules” and “HAP-particles can serve to block dentin tubules and contribute to their remineralization”. This can lead to two questions: 1. Can any microparticles that can adsorb on teeth reduce dentin hypersensitivity
Authors’ response:
We refer the reviewer to Figure 4 and the legend of Figure 4, as well as the previously published reviews on dentin hypersensitivity 8 of which are summarized in the supplement table (see references 8-15). Physically occluding dentin tubules with microparticles or molecules is a common goal (as stated in the second last paragraph of our introduction). Particles that have been found to occlude dentin include silica, Bioglass, strontium chloride combined with calcium carbonate, strontium acetate, and casein-phosphoprotein amorphous calcium phosphate. HAP, as shown in many in vitro and in vivo trials, is most suitable as a blocking agent because it has the same chemical structure as native dentin hydroxyapatite and has a better interaction with the dentin tubules.
- It is well known that saliva is rich in proteins. How does protein desorption kinetics influence particle adhesion? Recent work (Lyu, Biomacromolecules, 2022, 4709) shows that protein desorption changes with time. The protein adsorption should influence the particle deposition, but how? Does it affect the usage of oral care products, usually twice daily, to reduce dentin hypersensitivity?
Authors’ response:
The reviewer poses interesting mechanistic questions, the answers to which have not been obtained through careful studies of salivary protein interactions with hydroxyapatite, both at the level of the tooth surface and through interaction with synthetic hydroxyapatite in suspension. While some details have been studied (e.g., by Heller et al, J. Dent. Res. 2007 https://pubmed.ncbi.nlm.nih.gov/27879420/), it is likely that synthetic HAP particles bind salivary proteins. This may be one of the mechanisms through which they reduce bacterial adherence to the enamel surfaces as reviewed by Limeback et al. in 2021 (reference 18 in the review). Salivary protein desorption with time means that the biomimetic HAP binding to the dentin surface, if it relies in part in salivary protein binding, will affect the time that the HAP particles adhere to the surface. What the in vitro, in situ and clinical studies on remineralization have shown is that HAP gets incorporated into the mineral phase of dentin. Whether salivary protein desorption is a critical step in the incorporation of HAP into the mineral phase has not been studied at the molecular level but the clinical trials clearly show long term (months) of DH relief. Since this mechanistic discussion remains hypothetical until scientific studies have been conducted, we did not add to the discussion section on this subject.
- I would appreciate it if the authors could provide more discussions on the effect of pH on HAP deposition. Why and how does the dissociation of HAP contribute to the remineralization of the dentin surface?
Authors’ response:
We have mentioned the various ways that HAP remineralizes dentin. In the discussion we discussed how low pH dissociates HAP into calcium and phosphate ions (lines 934-937, and Figure 5). Remineralization of mineral crystals in the dentin need these ions to grow in size (in addition to having large HAP molecules trapped or adhering to the mineralize phase of dentin). In a low pH oral environment (when dental plaque metabolizes sucrose, or when a weak acid is consumed such as citric acid or acetic acid) more calcium and phosphate ions are present from the dissociation of the HAP particles than would normally be in saliva. This encourages remineralization of mineral phase of dentin. The study by Cieplik et al. in 2020 (reference 70) demonstrated that mechanism and it was mentioned in the review by Enax et al in 2019 reference 71.
We have altered this passage:
"In addition, studies have shown that lower plaque pH (or lower pH from dietary exposures) encourages dissociation of the HAP particles into calcium and phosphate ions [66] that penetrate further into the tubules and help to remineralize them [71]."
to
"In addition, studies have shown that lower plaque pH (or lower pH from dietary exposures) encourages dissociation of the HAP particles into calcium and phosphate ions [66, 70, 71]. Existing HAP crystals will grow in the presence of excess calcium and phosphate ions. These are provided by the dissociation of HAP particles supplied by the oral care products exposed to lower pH. The calcium and phosphate ions diffuse further into the dentin tubules. The pH is higher in the tubules the further from the surface the ions diffuse, and eventually salivary buffers neutralize the weak acids. This encourages the mineral phase to remineralize, grow and occlude the tubules further than the physical obstructions provided by the HAP particles that are still intact."
Reviewer 2 Report
Revision of considerable interest for the dental sector which requires a minor revision before publication.
Abstract to highlight the data collected
Keywords few to add specific ones
Well described introduction.
Materials and methods incorporate and/or add to the discussions the other studies already published by the research group of Prof Scribante on the use of biomimetic hydroxyapatite in the management of white spots and mih where they also evaluate the reduction of sensitivity
Conclusion Add proactive action to reduce the incidence of injury
Author Response
Revision of considerable interest for the dental sector which requires a minor revision before publication.
Abstract to highlight the data collected
Authors’ response:
We believe that the abstract accurately reports the results of the systematic review and meta-analyses. Total word restriction of the abstract prevents us from adding more information.
Keywords few to add specific ones
Authors’ response:
We believe the reviewer is asking us to add more key words. This is a good suggestion.
We therefore have added the following key words: “dentin tubule”, “quality of life”, “pain”, “remineralization”
Well described introduction.
Authors’ response:
We appreciate that no changes are required for the introduction
Materials and methods incorporate and/or add to the discussions the other studies already published by the research group of Prof Scribante on the use of biomimetic hydroxyapatite in the management of white spots and mih where they also evaluate the reduction of sensitivity
Authors’ response:
Since the search words used in the systematic review could have missed a publication or two because the search words did not include authors, we found the paper of Prof. Scribante to which the reviewer was referring. This paper:
Butera A, Pascadopoli M, Pellegrini M, Trapani B, Gallo S, Radu M, Scribante A. Biomimetic hydroxyapatite paste for molar-incisor hypomineralization: A randomized clinical trial. Oral Dis. 2022 Sep 22.
was published after our search was completed.
We also found other publications that were published after the search which would have been included in the list of trials and which we had neglected to mention in the manuscript body (but were included in the reference list in the first version of the manuscript downloaded).
The following section has been added at the end of the discussion.
"4.4 Additional studies
After our search was completed 4 additional studies appeared in the literature that were not included in this qualitative or quantitative synthesis [78-81]. These studies were consistent with HAP reducing dentin hypersensitivity. In all, 48 trials have been published examining the effectiveness of HAP as a dentin desensitizing additive in oral care products."
Conclusion Add proactive action to reduce the incidence of injury
Authors’ response:
We presume that this reviewer would like us to mention that dental professionals can reduce dental pain with HAP, a form of dental injury.
We expanded the last sentence in the conclusion to this:
"Dental professionals can consider recommending hydroxyapatite-based oral care products as a primary strategy for the effective management of dentin hypersensitivity which providing their patients with immediate and long-lasting relief from dental pain caused by dentin hypersensitivity."
Reviewer 3 Report
In the review manuscript “Clinical evidence of biomimetic hydroxyapatite in oral care products for reducing dentin hypersensitivity: an updated systematic review and meta-analysis” the authors present interesting information obtained by conducting a systematic review and meta-analysis of the current evidence that HAP-containing oral care products reduce dentin hypersensitivity. This referee can recommend it for publication after some comments and questions are fixed and clarified in order for the manuscript to be suitable for publication in Biomimetics.
Title
How do the results of this review relate to the intention to exploit biologically inspired designs to develop innovative solutions for systems in engineering, technology and biomedicine?
Table 1
For interactions in dentine nano HAP mimicked biologically produced HAP mineralizing the tissue constituted mainly by collagen. Not all the formulations given in the table 1 specify the nanocrystalline feature for HAP. Please include some comments related with crystalline size and morphology in section 4.1.
Author Response
In the review manuscript “Clinical evidence of biomimetic hydroxyapatite in oral care products for reducing dentin hypersensitivity: an updated systematic review and meta-analysis” the authors present interesting information obtained by conducting a systematic review and meta-analysis of the current evidence that HAP-containing oral care products reduce dentin hypersensitivity. This referee can recommend it for publication after some comments and questions are fixed and clarified in order for the manuscript to be suitable for publication in Biomimetics.
Title
How do the results of this review relate to the intention to exploit biologically inspired designs to develop innovative solutions for systems in engineering, technology and biomedicine?
Authors’ response:
This review was selected for a special issue of Biomimetics.
https://www.mdpi.com/journal/biomimetics/special_issues/0T0L61BWL1
Specifically, the guest editors said this:
"This Special Issue aims to collect contributions from different laboratories working on biomimetic remineralization. Covering issues from proof-of-concept to (basics) clinical translation, it provides an updated view of the potential strategies that would allow the transfer of the biomimetic remineralization method into the dental clinic. The present collection of papers takes advantage of the open access format and is expected to provide a paradigm of the power of biomimetic approaches for discovering new and important research avenues and innovative solutions in nanotechnology and dentistry.
We believe that this initiative will fill an essential gap in biomimetic remineralization, and the clinical delivery system will stimulate enthusiastic contributions from leading experts in the field.
Dr. Hamid Nurrohman
Dr. Jinhui Tao
Guest Editors
We feel that this systematic review of dentin hypersensitivity treatment with biomimetic hydroxyapatite (i.e., an active ingredient that is inspired by the natural tooth mineral) meets the goals of this special issue.
Table 1
For interactions in dentine nano HAP mimicked biologically produced HAP mineralizing the tissue constituted mainly by collagen.
Authors’ response:
We are not sure if this is a statement or a question. The review may simply want us to mention that the biomimetic HAP is similar in chemistry to the natural HAP found in and around collagen fibre bundles of dentin.
We have expanded the last sentence in the second last paragraph of the Introduction to the following.
"Biomimetic HAP seems to be one of those agents that fulfills both roles, since it is very similar to the HAP crystals between and intertwined within the collagen fibre bundles of dentin."
Not all the formulations given in the table 1 specify the nanocrystalline feature for HAP. Please include some comments related with crystalline size and morphology in section 4.1.
Authors’ response:
This was a good suggestion and we have added this passage to section 4.1.
"The oral care products used in the studies listed in Table varied in their synthetic HAP particle sizes. HAP is synthesized in various processes, which leads to a variety of particle dimensions. Most particles are micrometer in size, and can sometimes be measured in nanometer widths, as shown in many SEM studies in vitro. With dentin tubule width in the µM range (Figure 5) the small dimensions of the nano- and micro-HAP particles explains how they can accumulate in the dentin tubules and eventually occlude them, reducing dentin hypersensitivity. Specific crystal morphology and size has not been studied in detail to determine what is the optimum dimensions for biomimetic HAP should be for dentin tubule occlusion and dentin adherence."